# The Determination of Some Microbiological and Chemical Features in Herby Cheese

**DOI:** 10.3390/foods8010023

**Published:** 2019-01-11

**Authors:** Kamil Ekici, Hayrettin Okut, Ozgur Isleyici, Yakup Can Sancak, Rabia Mehtap Tuncay

**Affiliations:** 1Department of Food Hygiene and Technology, Veterinary College, Yuzuncu Yıl University, Van 65080, Turkey; oisleyici@hotmail.com (O.I.); ycsancak@yyu.edu.tr (Y.C.S.); r.m.gunes@hotmail.com (R.M.T.); 2Departement of Preventive Medicine, School of Medical, University of Kansas, Kansas City, KS 66160, USA; hokut@kumc.edu

**Keywords:** biogenic amines, histamine, herby cheese, HPLC, decarboxylase enzymes

## Abstract

The objective of this study is to measure the amounts of biogenic amines, microbial counts, values of pH, titratable acidity, dry matter, and salt (%) in herby cheese, a very popular staple in the Turkish diet, and to evaluate the concentration of biogenic amines in terms of public health risks. A high-performance liquid chromatography (HPLC) method was used for the determination of eight biogenic amines in 100 herby cheeses sold in the local markets of Van. The bacterial load of the herby cheeses ranged between 4.0 and 8.90 log CFU/g for viable total aerobic mesophilic bacteria (TAMB), <1 and 7.0 log CFU/g for lactic bacteria (LAB), <1 and 6.08 log CFU/g for coliform bacteria, <1 and 5.81 log CFU/g for Enterobacteriaceae, <1 and 2.60 log CFU/g for *Staphylococcus aureus*, and 3.70 and 8.05 log CFU/g for yeasts and molds. The results obtained suggested significant changes in the pH, titratable acidity, dry matter, and salt contents of the examined herby cheese samples. The detection levels of biogenic amines in the samples ranged from <0.025 to 33.36 mg/kg for tryptamine, from <0.038 to 404.57 mg/kg for β-phenylethylamine, from 0.03 to 426.35 mg/kg for putrescine, from <0.039 to 1438.22 mg/kg for cadaverine, from <0.033 to 469 mg/kg for histamine, from <0.309 to 725.21 mg/kg for tyramine, from <0.114 to 1.70 mg/kg for spermidine, and from <0.109 to 1.88 mg/kg for spermine. As a result, these cheeses are fit for consumption in terms of the amounts of biogenic amines they contain.

## 1. Introduction

Biogenic amines (BAs) are organic bases with an aliphatic, aromatic, or heterocyclic structure, which have been found in many foods, such as fish products, cheese, wine, beer, and other fermented foods [1,2,3]. Biogenic amine accumulation in foods usually results from the decarboxylation of amino acids by enzymes of bacterial origin, which is associated with food hygiene and technology [3,4,5]. The term “biogenic amines” defines decarboxylation products such as histamine, serotonin, tyramine, phenylethylamine, tryptamine, and also aliphatic polyamines [6].

Numerous bacteria, both intentional and adventitious, have been reported as being capable of producing biogenic amines. These are *Escherichia*, *Enterobacter*, *Salmonella*, *Shigella*, *Clostridium perfringens*, *Streptococcus*, *Lactobacillus*, and *Leuconostoc* [7,8,9]. The main biogenic amine producers in cheese are Gram-positive bacteria, with LAB being the main histamine and tyramine producers [10]. *Leuconostoc mesenteroides* has a high potential to form tyramine or histamine in wine [11,12]. The presence of biogenic amines in food constitutes a potential public health concern due to their psychological and toxicological effects [13]. Biogenic amines may also be considered as carcinogens because they are able to react with nitrites to form potentially carcinogenic nitrosamines [14]. Consuming contaminated fish of the Scombridae family is the most common type of fish poisoning in Europe and worldwide, as it is consumed in large quantities (tuna, bonito, and mackerel), thereby causing a pseudoallergic poisoning known as the scombroid syndrome. This syndrome may be triggered by various species from the family, such as sea urchins, bluefish, herring, anchovies, sardines, and dolphin fish. Poisoning resulting from the ingestion of this fish accounts for 40% and up to 5% of all food poisoning cases reported in the United States [15]. Histamine poisoning is the most common and toxic form of poisoning. Histamine intoxication, also termed scombroid poisoning, is an important foodborne disease common all over the world [8]. The intake of foods with high concentrations of biogenic amines can cause migraines, headaches, gastric and intestinal problems, and pseudoallergic responses [9,16].

Cheese represents an ideal environment for biogenic amine production. Several factors may contribute to biogenic amine formation in cheese. The utilization of raw or pasteurized milk in cheesemaking, higher ripening temperature, excessive proteolysis, high pH, and low salt concentration may contribute to the ability of an organism to produce biogenic amines [17,18,19].

Herby cheese, which has a semihard texture and a salty taste, is produced in small family businesses for their needs and for commercial purposes in Van city. Van city is located by the shores of Lake Van, in the eastern part of Turkey, bordering Iran. In addition, it is produced in well-equipped factories. It is made from raw sheep milk in the eastern and southeastern parts of Turkey. If sheep milk is not available, a mixture of sheep and cow or sheep and goat milk can be used for cheesemaking. Herby cheese, named “Otlu peynir” in Turkish, is homemade in villages and by some small local producers for many years. Most people consume it as a part of almost every meal [20,21,22]. There has been a continuous demand for this cheese in recent years, and this may further increase in the future, since the market of herby cheese has been spreading to the big cities in the country [23].

This study was undertaken to determine the amounts of biogenic amines in herby cheese, since biogenic amines are important with regard to toxicological effects. Besides the content of biogenic amines, microbial counts, values of pH, titratable acidity, dry matter, and salt parameters were also measured to provide complementary information on the microbiological and biochemical features of herby cheese, focusing on hygiene and the consumer health aspects of herby cheese.

## 2. Materials and Methods

### 2.1. Sample Origin

In the present study, 100 samples of herby cheese, collected from various sales points in Van city center, were used in this study. The samples were placed in sterile jars and brought to the laboratory in a cooler (3–6 °C) and analyzed immediately. Assays were done on duplicate samples with the results being averaged.

### 2.2. Microbiological Analysis

In brief, 10 g of herby cheese samples were weighed in stomacher bags, and then 90 mL of sterile physiologic water with peptone (0.85% NaCl + 0.1% peptone) was added. Then, samples were homogenized in stomacher bags for 2 min. After homogenization, serial decimal dilutions were prepared until dilutions of 10^9^ CFU/mL were reached, and from these dilutions, Petri dishes were inoculated. To assess the viable total aerobic mesophilic bacteria (TAMB), plates of plate count agar (Oxoid CM325) were incubated for 48 h at 35 °C; for lactic acid bacteria (LAB), plates of de Man, Rogosa and Sharpe (MRS) agar (Oxoid CM 361) were incubated for two days at 35 °C; for yeasts and molds (YM), plates of potato dextrose agar (Oxoid CM139) were incubated for five days at 21 °C; for coliform bacteria, plates of violet red bile lactose agar (Oxoid CM 107) were incubated for 24 h at 35 °C; for Enterobacteriaceae, plates of violet red bile glucose agar (Oxoid CM 485) were incubated for 48 h at 32 °C, and for *S. aureus*, plates of Baird–Parker agar (Oxoid CM 275) were incubated for 48 h at 35 °C. After inoculation on Petri dishes, colonies formed were counted [24].

### 2.3. Chemical Analysis

Herby cheese samples were analyzed for titratable acidity, also known as lactic acid (LA%), and dry matter, salt (%), and pH according to the method described by Tekinsen et al. [25].

### 2.4. Biogenic Amine Analysis

Sample preparation and biogenic amine analysis were done according to the high-performance liquid chromatography (HPLC) method described by Eerola et al. [26].

#### 2.4.1. Sample Preparation and Homogenization Procedure

One hundred herby cheese samples were collected, where each one weighed approximately 750 g. Then, the herby cheese samples were sliced with a clean stainless steel knife and were grated and homogenized thoroughly, and from each cheese sample, a 2 g sample was weighed (to the nearest 0.001 g) and transferred into a plastic Falcon tubes, then homogenized with a metallic staff homogenizer tool (T-25 digital Ultra-Turrax from IKA^®^-Works, Inc., Wilmington, NC, USA) for about 2 min. The homogenization was done by adding 125 μL of an internal standard (1.7-diaminoheptane) Sigma-Aldrich Company, St. Louis, MO, USA) with 10 mL of 0.4 M perchloric acid (Merck, Darmstadt, Germany) to the sample. In the next step, the homogenized samples were centrifuged (1210× *g* for 10 min under 4 °C) by high-speed refrigerated centrifuge (Hitachi Koki Co., Ltd., Tokyo, Japan), and then the extraction solvents were transferred and filtered with filter paper (Schleicher and Schuell, 589 Black ribbon Ø 70 mm, Dassel, Germany) into a volumetric flask. The remaining (supernatant) part was again centrifuged with 10 mL of perchloric acid and filtered into the same volumetric flask, then supplemented to 25 mL with 0.4 M perchloric acid. An aliquot of 1 mL of the final extract was then used for analysis after derivatization, while the remaining volume was stored at 4 °C for no more than one week.

#### 2.4.2. Derivatization of Extracts and Standards

Eight aqueous standard solutions containing cadaverine dihydrochloride (purchased from Aldrich Company, Buchs, Switzerland), putrescine dihydrochloride, tyramine hydrochloride, histamine dihydrochloride, tryptamine hydrochloride, 2-phenylethylamine hydrochloride, spermidine, spermine, or 1.7-diaminoheptane (as the internal standard) (all purchased from Sigma-Aldrich, St. Louis, MO, USA) were obtained. The standards used were prepared by our group with materials purchased from the respective suppliers. The dansylated derivatives of the amines were formed by adding 1 mL of sample extract or standard solution to 200 μL of 2 N NaOH (Merck, Germany) and 300 μL of saturated NaHCO_3_ (Merck) solution and mixing by vortex (Heidolph D-91126, Reax top, Schwabach, Germany), and then 2 mL of dansyl chloride solution and 2 mg of dansyl chloride per mL in acetone (Sigma) were added and the solution was again vortexed. Fresh dansyl chloride solutions were prepared each time immediately before use. After shaking, samples were left in the incubator at 40 °C for 45 min. After the reaction time had passed, the residual dansyl chloride was removed by the addition of 100 μL of ammonia (Merck) 25% (*v*/*v*), followed by vortex mixing and holding for 30 min at room temperature. The derivatization was completed upon the addition of an ammonium acetate (Merck, Germany) and acetonitrile (Merck) mixture (1:1; *v*/*v*) and adjustment to 5 mL. Finally, the mixture was centrifuged (Hettich Zentrifugen, Werk Nr, Tuttlingen, Germany) at 1210× *g* for 5 min under 4 °C and the supernatant was filtered through 0.45-μm pore-size filters (Millipore Co., Bedford, MA, USA).

#### 2.4.3. Chromatographic Conditions

Two solvent reservoirs containing (A) ammonium acetate and (B) acetonitrile were used to separate all the amines with an HPLC elution program (Shimadzu, Kyoto, Japan). The gradient–elution system used 0.1 M ammonium acetate as solvent A and acetonitrile as solvent B. The gradient–elution program was started with 50% solvent B and ended at 90% solvent B after 25 min. The system was equilibrated for 10 min before the next analysis. The flow rate was 1.0 mL/min and the column temperature was 40 °C. A 20 μL sample was injected onto the column. The quantitative determinations were carried out by an internal standard (1.7-diaminoheptane) method using peak heights.

### 2.5. Statistical Analysis of Data

SAS version 9.4 (SAS Institute, Inc. Cary, NC, USA) [27] was used for all data analysis. PROC UNIVARIATE in SAS was used for the descriptive statistics for variables. The results were defined as the mean values ± standard error of the mean.

## 3. Results

Table 1, Table 2 and Table 3 show the results of the microbial counts (in log CFU/g), chemical results, and biogenic amine levels (mean ± SE (standard error of the mean), mg/kg wet weight) found for the herby cheese samples. Log_10_ transformations were applied on the microbiological data. Presumptive viable total aerobic mesophilic bacteria (TAMB), lactic bacteria (LAB), Enterobacteriaceae, coliform bacteria, *Staphylococcus aureus*, and yeast and mold counts were investigated as general microbiological quality parameters. As can be seen from Table 1, the bacterial load of herby cheese ranged between 4.0 and 8.90 log CFU/g for viable total aerobic mesophilic bacteria (TAMB), <10 and 7.0 log CFU/g for lactic bacteria (LAB), <10 and 6.08 log CFU/g for coliform bacteria, <10 and 5.81 log CFU/g for Enterobacteriaceae, <10 and 2.60 log CFU/g for *S. aureus*, and 3.7 and 8.05 log CFU/g for yeasts and molds.

## 4. Discussion

Various studies have been carried out on the microbiological quality of herby cheeses in Turkey; for example, Ozturk [28] determined the TMAB, coliform, *S. aureus*, and mould–yeast counts in herby cheese to be 7.14, 3.96, 3.29, and 3.48 log CFU/g, respectively. Isleyici and Akyuz [29] reported a herby cheese TMAB count of 7.82 log CFU/g, mould–yeast count of 5.81 log CFU/g, coliform bacteria count of 2.23 log CFU/g, staphylococci count of 3.93 log CFU/g, and LAB count of 8.08 log CFU/g. Sagun et al. [30] found the TMAB, coliform, LAB, and mould–yeast counts in herby cheese to have the mean values of 6.24 ± 0.66, 2.99 ± 2.27, 5.48 ± 0.61, and 4.60 ± 2.11 log CFU/g, respectively. Tekinsen [31] determined the TMAB, Enterobacteriaceae, coliform, *S. aureus*, and mould–yeast counts in herby cheese as being 8.53, 5.44, 4.61, 4.34, and 5.50 log CFU/g, respectively. Alemdar and Agaoglu [32] reported the TMAB and LAB counts in herby cheese as 8.45 and 8.61 log CFU/g, respectively. These diverse results from various researchers and the present study could be explained by the nonstandardized production of herby cheese and the sale of both ripened and unripened cheeses in different storage conditions in the market [33].

Today, society is increasingly aware of the importance of diet for health, and hence, any issue relating to food safety has a considerable impact on consumer behavior and official policy [16]. Among fermented food products, cheese is most commonly related with biogenic amines (mostly histamine, tyramine, cadaverine, and putrescine) intoxication [34,35]. Recently, the EFSA (European Food Safety Authority) Panel on Biological Hazards (BIOHAZ) conducted a qualitative risk assessment for biogenic amines (BAs) in fermented foods, and concluded that our present knowledge of their toxicity was limited and that further research was needed [36,37]. In normal circumstances, the human body is able to rapidly detoxify histamine and tyramine absorbed from foods through acetylation and oxidation mediated by the enzymes monoamine oxidase (MAO; EC 1.4.3.4), diamine oxidase (DAO; EC 1.4.3.6), and polyamine oxidase (PAO; EC 1.5.3.11) [16]. The ingestion of biogenic amine (BA)-rich food can cause adverse toxicological reactions and intoxications harmful to health [38]. In fact, the presence of these biogenic amines in food, especially in conjunction with other factors, such as the consumption of monoamine oxidase-inhibiting drugs, alcohol, and other food amines (e.g., spermine, spermidine, putrescine, and cadaverine), may cause food poisoning [39].

Regarding the content of biogenic amines in herby cheese, the highest level of cadaverine was observed to be 1438.22 mg/kg (Table 3). Standarová et al. [40] reported that Olomouc tvorogs contained, among other amines, the highest level of cadaverine, at up to 2413.00 mg/kg. According to Bonczar et al. [35], this amine is predominant in Harzer cheese and can occur at the level of 377.50 mg/kg, and Fiechter et al. [41] found this amine at the level of 1268.00 mg/kg in Harzer cheese. According to Andiç et al. [22], the levels of cadaverine ranged from not detected to 1844.50 mg/kg in herby cheese. Vale and Gloria [42] reported finding cadaverine at levels of up to 1110.00 mg/kg in Brazilian cheese. These results are higher than that found in this study and are less than that found by Andiç et al. [22]. According to the European Food Safety Authority [36], fresh cheeses can contain cadaverine at levels from 10.70 to 45.00 mg/kg and hard cheeses from 47.80 to 83.50 mg/kg. The levels of cadaverine and putrescine are usually considered to be indicators of contamination and also markers of the hygiene standards of the production process. The representatives of the Enterobacteriaceae family and *Pseudomonas* genus are regarded as sources of cadaverine and putrescine [43].

The amount of histamine found in the herby cheese samples tested in this paper ranged from <0.033 to 469.00 mg/kg (Table 3). The histamine content of some cheeses varies widely; for example, Budak et al. [44] noted that the histamine concentration reached the level of 265.50 mg/kg after 90 days ripening at 10 °C, and it was detected in an amount higher than those of other biogenic amines in the Turkish white cheese samples investigated. Sancak et al. [45] reported that the histamine amount in 47 herby cheeses ranged between 25.62 and 957.62 mg/kg. Andiç et al. [22] reported that histamine levels ranged from 0 to 681.50 mg/kg. Antila et al. [46] found the amount of histamine in Emmental cheeses matured for 3 and 6 months to be 12.20 mg/100 g and 17.50 mg/100 g, respectively. According to Madejska et al. [47], the highest amount of histamine, of 730.47 ± 20.01 mg/kg, was found in in Gorgonzola Piccante cheese stored for 42 days at room temperature. The amount of histamine was less than what was reported previously by Madejska et al. [47] in Gorgonzola Piccante cheese and was in agreement with those reported previously by Andiç et al. [22] for herby cheese. The formation of histamine throughout the ripening period in herby cheese was studied by Sagun et al. [48], who reported that the concentration of histamine was 21.90 mg/kg on the first day of ripening, which then gradually increased and reached 46.20 mg/kg on the 90th day. Although the toxicity of histamine to man is a controversial subject, ingestion of 70–1000 mg histamine will usually cause the clinical symptoms of intoxication [49]. Food and Drug Administration (FDA) has established a hazard action concentration for histamine in tuna fish of 50 mg histamine/100 g [9,50]. The amounts of histamine found in the present research are lower than those that lead to clinical symptoms, which are 70–1000 mg [49]. The histamine content found in 10 of the 100 (10%) cheese samples was found to be higher than 200 mg/kg. The prevalence of this amine was high, but the levels detected were low and below its toxic threshold (50 mg histamine/100 g).

The amount of β-phenylethylamine found in the herby cheeses ranged from <0.038 to 404.57 mg/kg of sample (Table 3). The phenylethylamine level was previously reported to be 3.77 mg/kg in feta cheese by Valsamaki et al. [51]. Andiç et al. [22] reported that phenylethylamine levels in herby cheese ranged between 0 and 100 mg/kg, and Bonczar et al. [35] reported the mean phenylethylamine level of 8.76 ± 6.85 mg/kg in Emmental cheese.

The level of tryptamine determined in the herby cheeses ranged from <0.025 to 33.36 mg/kg of sample (Table 3). Andiç et al. [22] described tryptamine levels in the range of not detected to 172.60 mg/kg in herby cheese. According to Bonczar et al. [35], the mean amount of tryptamine was 48.91 ± 19.99 mg/kg in Harzer cheese. This variability within the same type of cheeses could be attributed to differences in the manufacturing process, such as the type of milk used (sheep or cow), heat treatment of the milk (such as pasteurization), ripening time, microflora, and cheese mass, as discussed by Andiç et al. [22].

The amount of tyramine found in the herby cheeses ranged from <0.309 to 725.21 mg/kg of sample (Table 3). According to Andiç et al. [22], tyramine levels in herby cheese ranged between 18.00 and 1125.50 mg/kg, and according to Fiechter et al. [41], were 51.60 mg/100 g in Harzer cheese with caraway seeds. Nout [52] pointed out that the maximum allowable level of tyramine in foods should be in the range of 100–800 mg/kg, and Shalaby [14] and Valsamaki et al. [51] stated that the safe summed level of histamine, tyramine, putrescine, and cadaverine should not significantly exceed the higher dose of 900 mg/kg. The concentrations of tyramine found in our study were lower than these levels.

Putrescine levels ranged from 0.03 to 426.35 mg/kg in the herby cheeses (Table 3). Bonczar et al. [35] described the mean putrescine level of 281.33 ± 114.90 in Harzer cheese. The biogenic amine (BA) content of cheese can be extremely variable and depends on the type of cheese, the ripening time, the manufacturing process, and the microorganisms present. The production of biogenic amines in cheese has often been associated with non-starter lactic acid bacteria and *Enterobacteriaceae* [53], so it may be a toxicological risk associated with the consumption of raw milk cheese, especially for sensitive individuals [34].

The mean level of spermine found in the herby cheeses was 0.25 ± 0.03 mg/kg of sample (Table 3). According to Vale and Gloria [42], spermine levels in Prato cheese ranged between 0.07 and 0.90 mg/kg, and according to Komprda et al. [1], the mean was 0.2 ± 0.1 mg/kg in Dutch-type hard cheese. According to Spizzirri et al. [54], the mean level of spermine in Parmigiano Reggiano cheese was 36.7 ± 2.3 mg/kg, and Bonczar et al. [35] reported the mean spermine level of 5.70 ± 1.48 mg/kg in Harzer cheese. The presence of these amines in herby cheese should be considered as a consequence of poor hygienic milk quality [22].

The mean content of spermidine found in herby cheese was 0.17 ± 0.03 mg/kg of sample (Table 3). According to El-Zahar [34], the mean level of spermidine in Mish cheese was 4 ± 0.63 mg/100 g, and Komprda et al. [1] reported the mean as being 0.3 ± 0.1 mg/kg in Dutch-type hard cheese. Spizzirri et al. [54] described that mean spermidine levels in Parmigiano Reggiano cheese were 73.1 ± 1.5 mg/kg, and Bonczar et al. [35] determined the mean spermidine content of Harzer cheese as being 7.74 ± 1.09 mg/kg.

The risk of biogenic amine poisoning could be controlled by applying basic good manufacturing and hygiene practices associated with an appropriate hazard analysis critical control point (HACCP) system [55]. In evaluating the risks of foodstuffs to the consumer’s health, the safe daily biogenic amines (BAs) intake should be regarded as complex. Some substances, such as alcohol, can decrease the activity of enzymes that participate in the degradation of BAs in the human intestines [14]. Since cheeses are often served with alcoholic drinks such as beer or wine, even low concentrations of BAs in cheeses can cause adverse effects. Beer and wine often contain high amounts of BAs, which might intensify the negative impact of BAs in cheeses on human health [43,56].

The pH value is an important factor influencing amino acid decarboxylase activity, which is stronger in an acidic environment, with the optimum pH level being between 4.00 and 5.50 [9]. The pH values ranged from 4.49 to 6.64 in this paper (Table 2). According to Andic et al. [22], the pH of herby cheeses ranges from 4.03 to 6.09. Kavaz et al. [57] reported the mean pH value in herby cheeses as being 5.82 ± 0.07. According to Tarakci et al. [20], the pH values of herby cheeses range from 4.01 to 5.40. The ripened herby cheeses showed pH values close to being alkaline, which is caused by the change of lactic acid into carbon dioxide. Sagun et al. [30] reported the pH value of herby cheese of 4.59 ± 0.44 as mean. Tuncturk et al. [58] described the mean pH value of herby cheeses as being 5.32 ± 0.04. Higher levels of organic acids or lower pH in ripened herby cheeses is found to be the specific result for this kind of cheese.

The dry matter content ranged from 49.66 to 65.80% with the mean of 60.37 ± 0.39% in this study (Table 2). Andic et al. [22] found the mean dry matter content in herby cheese to be 54.3 ± 1.03%. Kavaz et al. [57] reported the mean dry matter content as being 58.15 ± 0.28 in herby cheese, and Tarakci et al. [20] determined that the dry matter content of herby cheese ranges from 50.54 to 66.05%, with the mean being 55.41 ± 4.454%. Isleyici [59] found the mean dry matter content in herby cheese to be 47.783 ± 5.06%. Tekinsen [31] reported that the dry matter content in herby cheese ranges from 29.10 to 61.57%. According to Tuncturk et al. [58], the mean dry matter content in herby cheese was 46.01 ± 0.09%.

Titratable acidity (TA), measured by in lactic acid (LA%), ranged from 1.00 to 3.16 with the mean of 2.01 ± 0.04 in this study (Table 2). Tarakci et al. [20] reported that the titratable acidity (LA%) of herby cheese ranges from 0.82 to 2.35 with the mean of 1.84 ± 0.374. Isleyici [59] found the mean TA in herby cheese to be 0.809 ± 0.333%, and Sagun et al. [30] reported the TA in herby cheese of 1.18 ± 0.21. Tekinsen [31] reported that the TA of herby cheese ranges from 0.184 to 1.757. According to Tuncturk et al. [58], the mean TA of herby cheese is 1.47 ± 0.05.

The salt levels (%) ranged from 5.85 to 11.70% with the mean of 8.64 ± 0.17% in this study (Table 2). Sagun et al. [30] described that the mean salt level in herby cheese was 5.14 ± 0.61%. Isleyici [59] found the mean salt level in herby cheese to be 5.69 ± 1.11%. Kavaz et al. [57] reported the mean salt level to be 3.63 ± 0.14% in herby cheese. Tarakci et al. [20] found that the salt level of herby cheese ranged from 4.80 to 9.07% with the mean of 6.64 ± 1.190%. According to Andic et al. [22], the mean salt level in herby cheese was 9.01 ± 0.4%. Tuncturk et al. [58] described that the mean salt level in herby cheese was 4.45 ± 0.17%. There has been great variability in the results obtained for the chemical composition of the examined herby cheeses. This variability depends on many factors, such as the fresh milk quality, technological processes applied (pasteurization, starter culture addition, time and temperature of thermomechanical curd treatment, salting), and time and temperature of ripening, among many others [60].

## 5. Conclusions

The current study provides valuable information on the bacterial, chemical, and biogenic amine content in herby cheese. Taking into account all the microbiological results, it could be concluded that herby cheese can be highly prone to contamination, particularly with Enterobacteriaceae, coliform, and *S. aureus*, which is probably related to poor hygiene during cheesemaking manipulations. The use of raw milk and various herbs in the making of herby cheeses may result in pathogen contamination. Nevertheless, it can be said that the herby cheeses analyzed presented remarkably low bacterial densities. During the manufacture of herby cheese, it has been deemed necessary and highly appropriate to take several measures aimed at reducing the production of biogenic amines, to ensure sanitary conditions in making herby cheese, and to use starter cultures formed by lactic acid bacteria with acidifying capacity. The amines were not detected in every sample, and there was high variability in the amine levels among the samples analyzed. Histamine was only found in ten (10%) herby cheese samples at levels of up to 200 mg/kg, which can be considered toxicologically significant. The levels of spermine and spermidine were low and did not exceed the value of 35 mg/kg [52]. The levels of tyramine were also low and did not exceed the value of 800 mg/kg that is regarded as safe for the consumer’s health. These cheeses are fit for consumption in terms of the amount of biogenic amines they contain. Moreover, susceptible individuals should be advised to consume cheeses with low biogenic amine contents. However, the handling of raw materials and production technology for herby cheeses are relatively primitive in Turkey. Particularly if the person is vulnerable (when the histamine detoxification mechanism is inhibited), biogenic amines in amounts much lower than those mentioned here may cause intoxications. For this reason, the sources and critical control points for biogenic amine formation during cheesemaking should be determined in order to limit amine formation and accumulation in herby cheese.

## Figures and Tables

**Table 1 foods-08-00023-t001:** Results of microbial counts (in log CFU/g) in herby cheese samples.

Microorganisms	N	Mean ± SE	Min.	Max.
**TAMB**	100	5.52 ± 0.10	4.00	8.90
**LAB**	100	3.70 ± 0.17	<10	7.00
**Coliform**	100	3.57 ± 0.11	<10	6.08
**Enterobacteriaceae**	100	3.42 ± 0.13	<10	5.81
***S. aureus***	100	0.11 ± 0.04	<100	2.60
**YM**	100	5.31 ± 0.07	3.70	8.05

SE: standard error of the mean, TAMB: total mesophilic aerobic microorganisms, LAB: lactic acid bacteria, YM: yeasts and molds, N: number of analyzed samples.

**Table 2 foods-08-00023-t002:** Results of chemical results found in herby cheese samples.

Chemical Features	N	Mean ± SE	Min.	Max.
**Salt**	100	8.64 ± 0.17 (%)	5.85	11.70
**LA**	100	2.01 ± 0.04 (%)	1.00	3.16
**pH**	100	5.47 ± 0.04	4.49	6.64
**DM**	100	60.37 ± 0.39 (%)	49.66	65.80

LA%: lactic acid, DM: dry matter, N: number of analyzed samples.

**Table 3 foods-08-00023-t003:** Results of biogenic amine levels (mean ± SE, mg/kg wet weight) in herby cheese samples.

Biogenic Amines	N	Mean ± SE	Min.	Max.	LOD
**TR**	100	2.24 ± 0.47	ND	33.36	0.025
**PEA**	100	14.81 ± 4.53	ND	404.57	0.038
**PUT**	100	44.71 ± 8.06	0.03	426.35	0.028
**CAD**	100	98.42 ± 23.47	ND	1438.22	0.039
**HI**	100	54.20 ± 10.08	ND	469.00	0.033
**TY**	100	103.18 ± 17.10	ND	725.21	0.309
**SPD**	100	0.17 ± 0.03	ND	1.70	0.114
**SPM**	100	0.25 ± 0.03	ND	1.88	0.109

TR: tryptamine, PEA: β-phenylethylamine, PUT: putrescine, CAD: cadaverine, HI: histamine, TY: tyramine, SPD: spermidine, SPM: spermine, N: number of analyzed samples, ND: not detected.

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
