# Peer review of "The Determination of Some Microbiological and Chemical Features in Herby Cheese"

_foods, 2019, doi:10.3390/foods8010023_

Reviewer 1 Report

This manuscript requires additional revision before acceptance. The authors must address the following:

2.4.1. Sample Preparation and Homogenization Procedure - Line 101:

"Some parts of the 100 herby cheese (2 g of the sample) were randomly chosen"

This is inadequate information on sample preparation - How were sample weights determined? - To nearest 0.01g? or to nearest 0.1g?

Related to above question is the authors apparent lack of awareness of significant figures - Throughout the manuscript this is a problem:

For example some levels in Table 3 have 6 significant figures! This not possible after weighing out 2 g samples. As an example would the authors think it is realistic to report a sample weight as 2.0001 g ? - That is what would be required to report 6 significant figures in their table. Assuming they weighed samples to nearest 0.01g the most signficant figures possible is actually just 3 and if they weighed 2 g to nearest 0.1g it drops to 2 significant figures. The authors need to correct the above problem throughout the manuscript.

Regarding the following query and response:

Query No: AQ4
Can the authors comment on the methodology used to detect the amines? How does this method compare to the officially recognized method of

Duflos et al used in the EU? The authors should comment on this.

Response;
"The method applied to this study is commonly used by many rernowned researchers"

Reviewer Comment - The author's response above  is inadequate. It is not a question of how renowned the researchers are. For the reader of this manuscript the question will be 1) whether the method has been validated to same level as the EU method and 2) if the methods are equivalent enought that those using the official EU method would be able to reproduce these results and generally 3) how does the method used differ from the EU method? The authors need to address this adequately.

Author Response

Dear Reviewer 1 of the manuscript;;

 I have read and evaluated your comments about the manuscript.

 Now,i'm sending the answers to you i've prepared.

  My best wishes, Merry Christmas!

 Dr. Kamil Ekici

Response to Reviewer 1 Comments
 Point 1: 2.4.1. Sample Preparation and Homogenization Procedure - Line 101:

"Some parts of the 100 herby cheese (2 g of the sample) were randomly chosen"

This is inadequate information on sample preparation - How were sample weights determined? - To nearest 0.01g? or to nearest 0.1g?

Related to above question is the authors apparent lack of awareness of significant figures - Throughout the manuscript this is a problem:

For example some levels in Table 3 have 6 significant figures! This not possible after weighing out 2 g samples. As an example would the authors think it is realistic to report a sample weight as 2.0001 g ? - That is what would be required to report 6 significant figures in their table. Assuming they weighed samples to nearest 0.01g the most signficant figures possible is actually just 3 and if they weighed 2 g to nearest 0.1g it drops to 2 significant figures. The authors need to correct the above problem throughout the manuscript.

Response 1:

I think that, there is a confusing notion.

  No, we didn’t take two grams (2g) from one hundred Herby cheese.

 We have collected one hundred Herby cheese samples where each one weighed approximately 750 g. Then Herby cheese samples were sliced with a clean stainless steel knife and were grated and homogenised thoroughly, and from each cheese sample were weighed 2 g (to nearest 0.001g) from this Herby cheese and transferred into Falcon plastic tubes then homogenized with a metallic staff homogenizer tools .

I have indicated this correction on the text with green colour.

Point 2: Regarding the following query and response:

Query No: AQ4

Can the authors comment on the methodology used to detect the amines? How does this method compare to the officially recognized method of Duflos et al used in the EU? The authors should comment on this.

Response;
"The method applied to this study is commonly used by many rernowned researchers"

Reviewer Comment - The author's response above  is inadequate. It is not a question of how renowned the researchers are. For the reader of this manuscript the question will be 1) whether the method has been validated to same level as the EU method and 2) if the methods are equivalent enought that those using the official EU method would be able to reproduce these results and generally 3) how does the method used differ from the EU method? The authors need to address this adequately.

Response 2:

 There are many methods used in biogenic amine analysis such as Colorimetric methods, Thin layer chromatography (TLC) methods, Enzymatic methods, Immuno-Enzymatic methods, Flow injection analysis–FIA, Fluorimetric method for histamine, Gas Chromatography (GC) methods, High performance thin layer chromatography (HPTLC) methods, High Performance liquid chromatography (HPLC) methods.

We used in this manuscript High Performance liquid chromatography (HPLC) method described by Eerola et al. [1993].

Eerola, S.; Hinkkanen, R.; Lindfors, E.; Hirvi, T. Liquid chromatographic determination of biogenic amines in dry sausages. J. AOAC. Int. 1993, 76, 3, 575–577.

I think that here it is obvious that you are only familiar with and referring to the EU method, the administered method is a valid and a scientific method.

Reviewer 2 Report

There are still some aspects in the manuscript that must be improved, among which is the drafting of the manuscript, which in some points is unclear. The manuscript should be reviewed by an English speaker to ensure the correct wording of some sections.

Abstract Ln 24-27: The sentence included as a conclusion is confusing and contradictory. It implies that there is no risk, but in turn it can cause intoxication. The drafted should be reviewed.

Ln 46-49. Write "family" in lowercase. The fragment also contains several grammatical errors that should be corrected.

Ln 61-62: Review the structure of the Phrase: "... the Eastern part of Turkey borders Iran."

Ln 106: if the information is included as "g" it is not necessary to include it as "rpm"

Ln 114-132: When the name of a brand is mentioned for the first time it is necessary to include the place and country of origin, but this does not need to be repeated in subsequent citations, where it is sufficient to mention the name of the brand.

Ln 142-142: Besides mentioning the program used, it is necessary to mention the kind of statistical tests used in the statistical analysis (ANOVA, Tukey ...). It is also necessary to include the number of samples used in the comparisons. SAS is a trademark, so other information should be included, such as the version and the place / country of origin of the product.

Ln 151-153 and throughout the manuscript: the units (Log CFU / g) should be placed just after each numerical value.

Title of table 1: put the units in parentheses

Title of table 2: "Results of chemical results ..." is redundant. Change "acidite" to "acidity" in the footer of the table. The units of some parameters are missing.

Table 3. What is LOD? It would be better to put a “nd” (non detected), including the limit of detection at the footer of the table instead of BDL.

Ln 198. If this is a type of cheese, put the first letter in uppercase

The response given to the AQ24 query is not satisfactory. It should not be difficult to analyse and mention if there is any relationship between the parameters observed and the levels of biogenic amines, even if the result of the comparison does not reflect any correlation.

314-317. The included text is still confusing and contain some grammatical errors. It must be revised.

Ln 319. Reference [52] should be included to indicate the source of the level of histamine considered toxic in the conclusions.

Ln 329-330: Why do you come to this conclusion? Is there any data that indicates that the decarboxylase microbiota comes from the spices?

Author Response

Dear Reviewer 2 of the manuscript;;

 I have read and evaluated your comments about the manuscript.

 Now,i'm sending the answers to you i've prepared.

  My best wishes, Merry Christmas!

Dr. Kamil Ekici

Response to Reviewer 2 Comments

 Point 1: There are still some aspects in the manuscript that must be improved, among which is the drafting of the manuscript, which in some points is unclear. The manuscript should be reviewed by an English speaker to ensure the correct wording of some sections.

 Response 1:

Manuscript has been corrected by my dear English teacher Marion Kelly from Swan Training Institute and Trinity College Dublin, Ireland.

 Point 2: Abstract Ln 24-27: The sentence included as a conclusion is confusing and contradictory. It implies that there is no risk, but in turn it can cause intoxication. The drafted should be reviewed.

 Response 2:

Yes, I have deleted this sentence from text.

“If a person is vulnerable (when the histamine detoxification mechanism is inhibited) histamine in amounts much lower than those mentioned may cause intoxications.

 Point 3: Ln 46-49. Write "family" in lowercase. The fragment also contains several grammatical errors that should be corrected.

Response 3:

Yes, I have wrote “family” in lowercase on the text.

 Point 4: Ln 61-62: Review the structure of the Phrase: "... the Eastern part of Turkey borders Iran."

Response 4:

Yes, I have changed the structure of the Phrase Van city is located by the shores of Lake Van, in the Eastern part of Turkey, bordering Iran.

Point 5: Ln 106: if the information is included as "g" it is not necessary to include it as "rpm"

Response 5:

Yes, I have deleted "rpm" from text

Point 6: Ln 114-132: When the name of a brand is mentioned for the first time it is necessary to include the place and country of origin, but this does not need to be repeated in subsequent citations, where it is sufficient to mention the name of the brand.

Response 6:

Yes, I have changed on the text.

Point 7: Ln 142-142: Besides mentioning the program used, it is necessary to mention the kind of statistical tests used in the statistical analysis (ANOVA, Tukey ...). It is also necessary to include the number of samples used in the comparisons. SAS is a trademark, so other information should be included, such as the version and the place / country of origin of the product.

Response 7:

Handling of raw materials and production technology for Herby cheese is relatively primitive in Turkey. We have collected the cheese samples which is homemade. No brand. We analysed the cheeses produced at home and sold in the public market in Van city. And we used PROC UNIVARIATE for statistical analysis.

Point 8: Ln 151-153 and throughout the manuscript: the units (Log CFU / g) should be placed just after each numerical value.

Response 8:

Yes, I have changed on the text.

Point 9: Title of table 1: put the units in parentheses

Response 9:

Yes, I have changed on the Table 1.

Point 10: Title of table 2: "Results of chemical results ..." is redundant. Change "acidite" to "acidity" in the footer of the table. The units of some parameters are missing.

Response 10:

Yes, I have corrected and added units on the Table 2 in the footer.

Point 11: Table 3. What is LOD? It would be better to put a “nd” (non detected), including the limit of detection at the footer of the table instead of BDL.

Response 11:

Yes, I have changed on the Table 3

Point 12: Ln 198. If this is a type of cheese, put the first letter in uppercase

Response 12:

Yes, I have changed on the text.

 Point 13: The response given to the AQ24 query is not satisfactory. It should not be difficult to analyse and mention if there is any relationship between the parameters observed and the levels of biogenic amines, even if the result of the comparison does not reflect any correlation.

Response 13:

 We did not conduct corelation analysis in this manuscript.

Point 14: 314-317. The included text is still confusing and contain some grammatical errors. It must be revised.

Response 14:

Manuscript has been corrected by my dear English teacher Marion Kelly from Swan Training Institute and Trinity College Dublin, Ireland.

Point 15: Ln 319. Reference [52] should be included to indicate the source of the level of histamine considered toxic in the conclusions.

Response 15:

Yes, i have included Reference [52] in the conclusions.

Point 16: Ln 329-330: Why do you come to this conclusion? Is there any data that indicates that the decarboxylase microbiota comes from the spices?

Response 16:

 We think that, but I have deleted this sentence from text

“This study also suggests that the herbs used for Herby cheese should be examined for the decarboxylating microbes in future studies”.

 This manuscript is a resubmission of an earlier submission. The following is a list of the peer review reports and author responses from that submission.

Round  1

Reviewer 1 Report

General comments.

The work describes the levels of biogenic amines present in samples of Herby cheese, a typical Turkish cheese variety. Cheese samples are also analysed for other parameters, both microbiological and chemical. The work is interesting because it provides data on the concentrations of biogenic amines in this type of products, although, as described in the discussion, there are several previous publications in this regard, so the authors should highlight where the originality of this work resides. Despite the high number of parameters analysed that could have an influence on the formation of BAs, no type of relationship has been determined that helps to explain the reason for the levels of amines found, simply describing the values obtained. The conclusions about the risk that the consumption of this cheese type can implies must also be explained in greater detail.

 Ln 15 “was” is missed in the sentence

Ln 13 and 16: “Cheese” or “cheese”?

Ln 17-19: Please, place the units (Log CFU/g) after each set of data, as you did for biogenic amine levels. The same for Ln 133-136 of “Results”

Ln 19: Write S. aureus in full

Where are your conclusions in the abstract?

Ln 64: delete “and”

Ln 16 and 70: please, describe a bit the position of Van in Turkey (south, west…?)

Ln 76: “than” or “then”?

Ln 78: “Petri”

Ln 78-84: Incubation conditions (Time/Temp)?

Ln 93: “Falcon”

Ln 96: What’s the brand of internal standard?

Ln 104-115: Where are the standards? Please, give the name and references of the biogenic amines you analysed. Some data concerning the recovery and the detection limit of the method would be also appreciated.

Ln 114: Please, give the centrifugation conditions in g instead of rpm

Ln 124: What kind of statistical analysis did you perform?

Ln 139: “Log”

Ln 132 (vs Ln 19, 80…): Please follow always the same language style (“British” or “US” English): “mold” or “mould”

Ln 148-150: this sentence is unnecessary

Ln 178,179, 180 and all over the manuscript… Use always the same number of decimals

Ln 180-181: harzer? Is that a type of cheese?

Ln 166 and 185: you must define what EFSA is when you cite it the first time in Ln 166

Ln 181-184; 194-202…..: considering that there are previous references about Herby cheese, the comparison with other cheese types is not necessary although you define clearly what these cheeses have in common. Consider what you say in Ln 219-221.

Ln 206-210: The reference Henry (1960) is not the most suitable to set the levels of risk of histamine. Please, take into consideration the EFSA report (ref 36)

Ln 230, 254, 255…: Why not defining the abbreviation BAs for “biogenic amines” the first time you cite them in Ln 31 and use this abbreviation along the manuscript?

Ln 256-288: If you consider that these factors can be influential in the Bas levels of Herby cheese some discussion comparing the correlation between these factor and the levels of Bas found would be appreciated, otherwise the merely description of the values is worthless for the main objective of your work.

Ln 297-298: This sentence seems incomplete.

Please, revise your conclusions concerning the risk of BA content in your Herby cheese samples. Why 200 mg/kg of histamine are toxicologically significant but 800 mg/kg of tyramine not? What’s the estimated intake of this cheese? As this is one of the main objectives of your work you may explain very well how you achieved these conclusions.

Reviewer 2 Report

The following text from line 21 in the abstract brings up the question of what level is "undetectable"

"The detection levels of biogenic amines in the samples ranged from undetectable to 33.36 mg/kg for tryptamine, from undetectable to 404.57 mg/kg for  beta-phenylethylamine, from 0.03 to 426.35 mg/kg for putrescine, from undetectable to 1438.22 mg/kg for cadaverine, from undetectable to 469 mg/kg for histamine, from undetectable to 725.21 mg/kg for tyramine, from undetectable to 1.70 mg/kg for spermidine, and from undetectable to

1.88 mg/kg for spermine"

The authors need to explecitly state what "undetectable" means. For this they need to state the LOD and LOQ of the method they are employing, and how these were determined. Although the phrase "umdetectable" appears throughout the manuscript, nowhere in the paper do they define these values and readers will wonder what these values are.

Further, and througout the manuscript, rather than use the word "undetectable" it would be more useful to say < x mg/kg, where x is the LOD for that particular biogenic amine. For example if the LOD for detecting histamine was, say, 1 mg/kg, the statement above for histamine would read: "..from < 1 mg/kg to 469 mg/kg for histamine,.." etc

Regarding the introduction, scombroid poisoning (or histamine poisoning) should be discussed in more detail since histamine is the only biogenic amine, among all of the others mentioned here, for which we have "advisory levels." Also, the authors should cite more contemporary and complete reveiws of histamine and scombroid poisoning than available from references 8 and 15.

Can the authors comment on the methodology used to detect the amines? How does this method compare to the officially recognized method of bDuflos et al used in the EU? The authors should comment on this.

Another area the authors need to fix is their use (or lack of use) of units. For example they state numbers obtained for TMAB but do not define TMAB (trimethylamine base?) and also do not state the units TMAB is stated in (is it mg/kg?)